# Plant Toxins as Potential Alternatives to Botulinum Toxin for Eye-Movement Disorder Therapy

Massimo Bortolotti [ID], Andrea Zanello [ID], Lorenzo Serra [ID], Francesco Biscotti [ID], Letizia Polito [†][ID] and Andrea Bolognesi *,[†][ID]

Department of Medical and Surgical Sciences—DIMEC, General Pathology Section, Alma Mater Studiorum—University of Bologna, 40126 Bologna, Italy
* Correspondence: andrea.bolognesi@unibo.it
† These authors contributed equally to this work.

**Abstract:** The most successful alternative to traditional surgery for ocular muscle spasm treatment is the intramuscular injection of botulinum toxin (BTX), which allows the maintenance of the muscle dynamics and the absence of scars. However, the main BTX disadvantage is its nonpermanent effect. A possible way for overcoming this obstacle could be represented by the enzymatic surgery using plant toxins known as ribosome-inactivating proteins (RIPs). In this paper, two highly toxic RIPs, namely, ricin and stenodactylin, were considered in a preliminary study for their possible use in the treatment of strabismus and oculofacial dystonias, as alternatives to BTX. Both RIPs showed a strong cytotoxic effect against rhabdomyosarcoma cell lines and myotube differentiated cells, with stenodactylin being about 10-fold more toxic than ricin. Moreover, stenodactylin showed a much higher cytotoxicity on myoblasts than on rhabdomyosarcoma cells. In our experimental conditions, stenodactylin did not damage conjunctival cells. Despite the limitations due to in vitro experiments, our data show that the high cytotoxicity of stenodactylin allows the use of a very low dose and, consequently, of very low injection volumes. This can represent a great advantage in the case of in vivo locoregional treatment. Furthermore, it is possible to modulate the chemoablation of myocytes while destroying myoblasts, thus reducing regenerative phenomena. The risk of cytotoxicity to surrounding tissues would be strongly reduced by the low injected volume and the relative resistance of conjunctival cells. In conclusion, our data suggest that stenodactylin and ricin could represent potential candidates to substitute BTX in ocular dystonia therapy.

**Keywords:** cytotoxicity; enzymatic surgery; myoblasts; ribosome-inactivating proteins; ricin; stenodactylin; strabismus



## 1. Introduction

A number of neurologic disorders are characterized by disabling involuntary muscular spasms that cause, when eye muscles are involved, strabismus and oculofacial dystonias. The most used treatment for ocular muscle spasm is the intramuscular injection of botulinum toxin type A (BTX) [1,2] in order to induce muscle relaxation. The main advantage of BTX treatment, with respect to the traditional surgery, is the maintaining of normal muscle dynamics that are often compromised by surgery. Furthermore, surgery may induce scar formation and disruption of relationships between muscle and soft-tissue pulleys, resulting in altered extraocular muscle function [3,4]. However, the therapeutic effect of BTX is not permanent, as the BTX paralytic effects vanish over time and patients typically require additional injections. In addition, the onset of a BTX resistance mechanism is common and includes collateral sprouts of denervated motor nerve terminals and formation of toxin-neutralizing antibodies [5,6]. As a result, higher and more frequent doses of BTX are necessary, thus increasing the risk of unwanted side effects including, although rare, eyelid ulceration. Eventually, some patients become refractory to BTX treatment [7,8]. Therefore, new alternative treatments are needed.

Ribosome-inactivating proteins (RIPs) are plant toxins widely distributed in the plant kingdom, and several RIP-producing plants have been used for centuries in traditional and folk medicines [9]. RIPs possess N-glycosylase activity (EC 3.2.2.22) [10] on several substrates, such as rRNA, DNA, viral nucleic acids, tRNA and polyA sequences [11–13], thus irreversibly damaging different substrates with the consequent arrest of protein synthesis and the induction of apoptosis [14] and of other cell-death pathways [15,16]. RIPs are classically divided into two main groups: type 1 RIPs, consisting of a single polypeptide chain with enzymatic activity, and type 2 RIPs, in which an A chain, similar to the type 1 RIPs, is linked to a B chain with the properties of a lectin specific for galactose, galactosamine or sialic acid [17].

RIPs can be linked to appropriate carriers to obtain conjugates that are specifically toxic to target cells. These conjugates, named immunotoxins, have been used for experimental therapies against various disease-causing cell targets. The best results were reported in hematological malignancies, where the efficacy of these hybrid molecules was particularly promising in controlling unwanted immune response [18,19]. Among the various possible uses, immunotoxins were also proposed for the locoregional treatment of strabismus, both alone [20,21] and in combination with BTX [22].

Ricin, purified from *Ricinus communis* L. seeds, is the best-known type 2 RIP, and its A chain is one of the most used toxic moieties in the construction of immunotoxins [23].

Stenodactylin, purified from the caudex of *Adenia stenodactyla* "Harms" [24,25], is the most potent type 2 RIP known to date, exhibiting very high toxicity against cells and animals. The high cytotoxicity of this protein depends on its high affinity for plasma membrane glycoproteins, efficient cell binding, endocytosis and intracellular routing, together with good resistance to proteolysis [26]. Moreover, stenodactylin was found to trigger multiple cell-death pathways [27], making it a good candidate for enzymatic surgery, an innovative and efficient strategy for the elimination of unwanted cells.

In this work, we investigated the in vitro effects of ricin and stenodactylin with the aim to develop an innovative experimental procedure for myocyte ablation in strabismus and other eye-movement disorders.

## 2. Results

### 2.1. Effect of Stenodactylin and Ricin on TE671 and RD/18 Protein Synthesis

Protein synthesis inhibition assay was performed on two undifferentiated rhabdomyosarcoma cell lines, namely, TE671 and RD/18. Stenodactylin and ricin inhibited L-[4,5-$^3$H]leucine incorporation in a dose-dependent manner (Figure 1a). The concentrations inhibiting protein synthesis by 50% and 90% (IC$_{50}$ and IC$_{90}$, respectively) are reported in Figure 1c. In undifferentiated TE671 and RD/18 cells, stenodactylin showed protein synthesis inhibition at concentrations at least 1 logarithm lower than ricin. Stenodactylin was significantly more efficient than ricin ($p < 0.001$) at concentrations ranging from 1 to 10 pM. In both cell lines, stenodactylin strongly reduced protein synthesis, to less than 10% at a 10 pM concentration, whereas ricin at the same concentration showed a lower effect, reducing protein synthesis to 84% and 41% in TE671 and RD/18, respectively.

In order to evaluate RIP protein synthesis inhibition on muscle-like cells, TE671 and RD/18 were grown in the presence of horse serum to induce their differentiation toward a muscle-like phenotype. These differentiated cells were less sensitive to RIPs than undifferentiated cells, with IC$_{50}$ values from 1.7- to 6.1-fold higher (Figure 1c). Moreover, in this model, stenodactylin caused a higher inhibition of protein synthesis than ricin (Figure 1b).

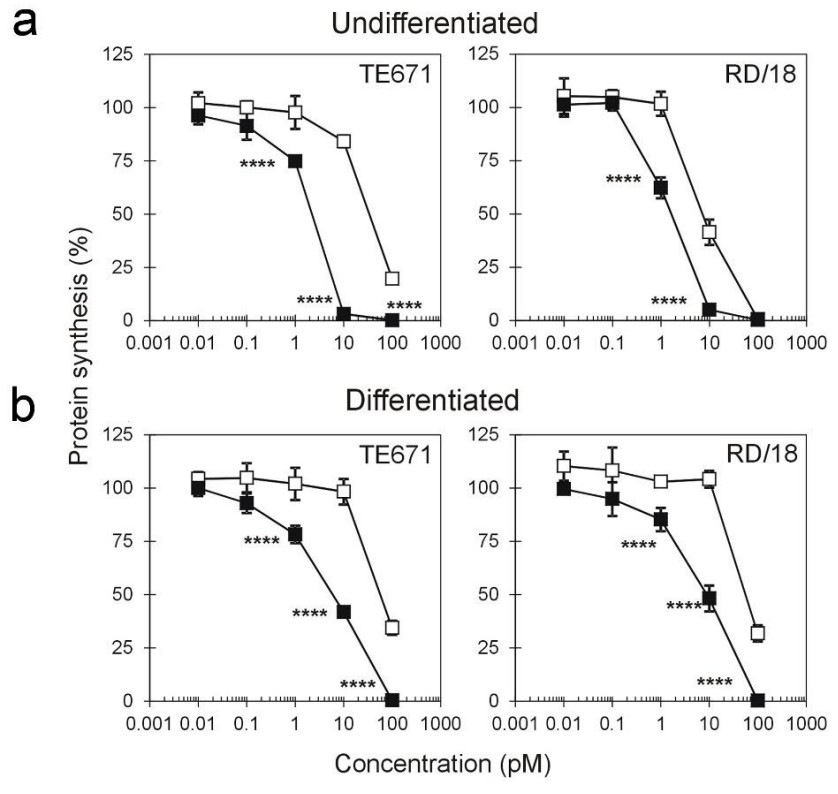

**Figure 1.** Protein synthesis inhibition assay on rhabdomyosarcoma cell lines. TE671 and RD/18 cells were treated with stenodactylin (black square) or ricin (white square). Undifferentiated (**a**) and differentiated (**b**) cells ($1 \times 10^4$/well) were seeded in 24-well plates in a total volume of 1 mL of complete medium containing scalar concentrations of RIPs. After 72 h of incubation and further 6 h with L-[4,5-$^3$H]leucine, the incorporated radioactivity was determined. Data were analyzed by the ANOVA/Bonferroni test (confidence range 95%; **** $p \leq 0.0001$, stenodactylin-*versus* ricin-treated cells). Results are means of three independent experiments, each performed in triplicate. SD (standard deviation) never exceeded 10%. Concentrations causing 50% (IC$_{50}$) and 90% (IC$_{90}$) of protein synthesis inhibition are reported (**c**).

## 2.2. Effect of Stenodactylin and Ricin on TE671 and RD/18 Cell Viability

RIPs were able to reduce cell viability both on undifferentiated and differentiated TE671 and RD/18 cells with a trend similar to that reported for protein synthesis inhibition curves (Figure 2a,b). In the results of these experiments, undifferentiated cells were more sensitive to RIPs than differentiated ones (Figure 2c). Viability was less than 10% in undifferentiated cells treated with stenodactylin at a 10 pM concentration in both cell lines, whereas the same effect was obtained in differentiated cells at a 100 pM stenodactylin concentration. Results showed that ricin was less efficient than stenodactylin in reducing

viability, indeed, the complete loss of viability was reached only in undifferentiated RD/18 cells at a 100 pM concentration.

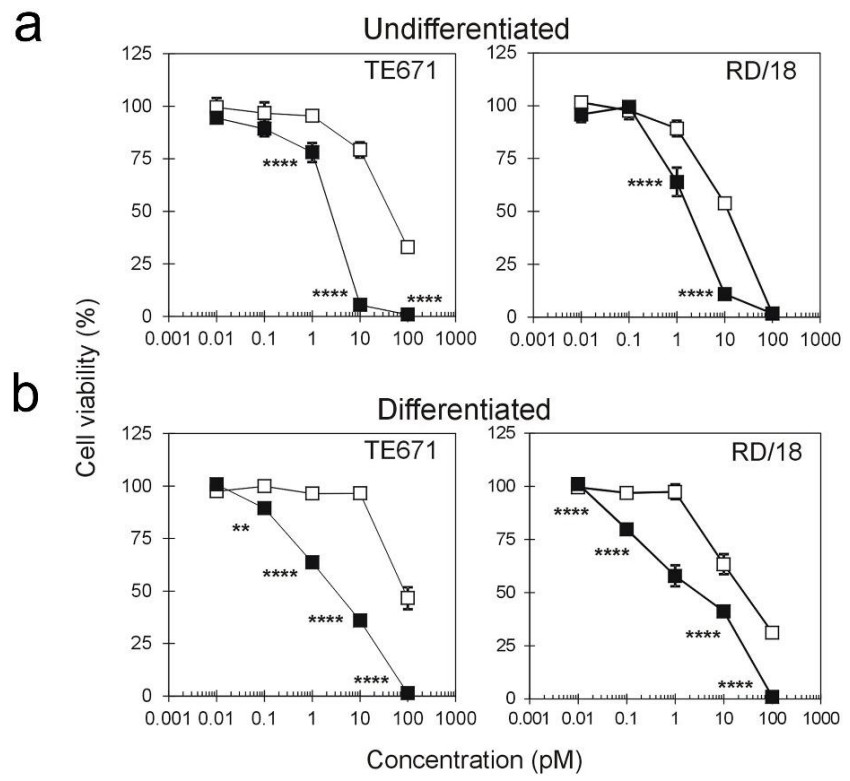

**Figure 2.** Cell viability assay on rhabdomyosarcoma cell lines. Undifferentiated (**a**) and differentiated (**b**) cells ($3 \times 10^3$/well) were seeded in 96-well plates in a total volume of 0.1 mL of complete medium containing scalar concentrations of stenodactylin (black square) or ricin (white square). After 72 h of incubation, MTS reduction was determined. Data were analyzed by the ANOVA/Bonferroni test (confidence range 95%; **** $p < 0.0001$; ** $p < 0.01$ stenodactylin-*versus* ricin-treated cells). Results are means of three different experiments, each performed in triplicate. SD never exceeded 10%. Concentrations causing 50% (LC$_{50}$) and 90% (LC$_{90}$) of MTS reduction inhibition are reported (**c**).

The table (panel c):

| Cell Line | RIP | LC$_{50}$ (pM) | | LC$_{90}$ (pM) | |
|---|---|---|---|---|---|
| | | Undiff. | Diff. | Undiff. | Diff. |
| TE671 | Ricin | 42.6 | 85.4 | >100 | >100 |
| | Stenodactylin | 2.4 | 3.2 | 8.6 | 55.9 |
| RD/18 | Ricin | 9.1 | 26.3 | 69.4 | >100 |
| | Stenodactylin | 1.8 | 3.1 | 12.4 | 59.5 |

### 2.3. Effect of Stenodactylin and Ricin on L6E9 Protein Synthesis and Cell Viability

The cytotoxic effect of RIPs was evaluated on the murine myoblast L6E9 cell line to investigate the possibility of obtaining permanent effects on cells implicated in regenerative phenomena. On L6E9 cells, stenodactylin showed a toxicity of about 3–4 logs higher than on rhabdomyosarcoma cell lines (Figure 3a–c). IC$_{50}$ and LC$_{50}$ (concentration that reduces the cell viability to 50%) were lower than the minimum tested concentration (0.01 pM). Instead, ricin showed a cytotoxic effect similar to or only 1 log lower than those reported for rhabdomyosarcoma cells.

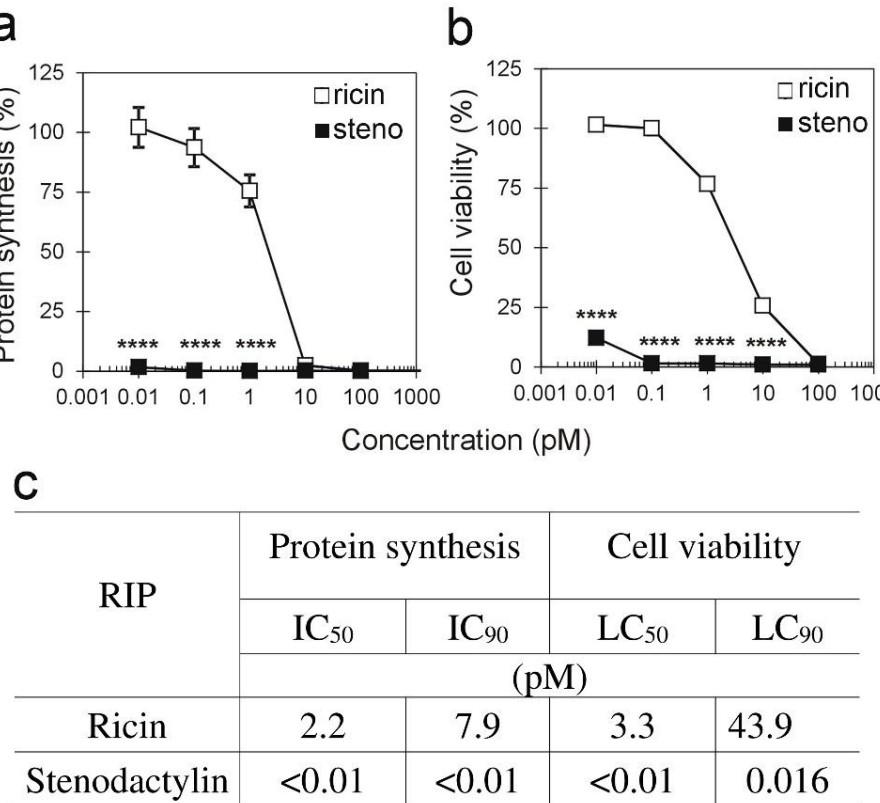

Figure 3. Protein synthesis inhibition and cell viability assay on L6E9 cells treated with stenodactylin or ricin. L6E9cells ($1 \times 10^4$/well) were seeded in 24-well plates in a total volume of 1 mL of complete medium containing scalar concentrations of stenodactylin (black squares) or ricin (white squares). After 72 h of incubation and further 6 h with L-[4,5-$^3$H]leucine, the incorporated radioactivity was determined (**a**). L6E9cells ($3 \times 10^3$/well) were seeded in 96-well plates in a total volume of 0.1 mL of complete medium containing scalar concentrations of stenodactylin or ricin. After 72 h of incubation, MTS reduction was determined (**b**). Data were analyzed by the ANOVA/Bonferroni test (confidence range 95%; **** $p < 0.0001$; stenodactylin- *versus* ricin-treated cells). Results are means of three different experiments, each performed in triplicate. SD never exceeded 10%. Concentrations causing 50% ($LC_{50}$) and 90% ($LC_{90}$) of MTS reduction inhibition and concentrations causing 50% ($IC_{50}$) and 90% ($IC_{90}$) of protein synthesis inhibition are reported (**c**).

## 2.4. Evaluation of Apoptosis Induced by Stenodactylin and Ricin

To evaluate the ability of RIPs to induce apoptosis, cell morphology, Annexin V/Propidium Iodide staining and caspase 3/7 activity were assayed. TE671 cells treated with stenodactylin for 24 h showed the typical aspect of early or late apoptotic cells, such as cell contraction and cytoplasmic vacuolization, loss of adhesion, rounded shape, great reduction of size, nuclear condensation and fragmentation (Figure 4a, upper panels). No necrotic cells were observed (Figure 4a, lower panels).

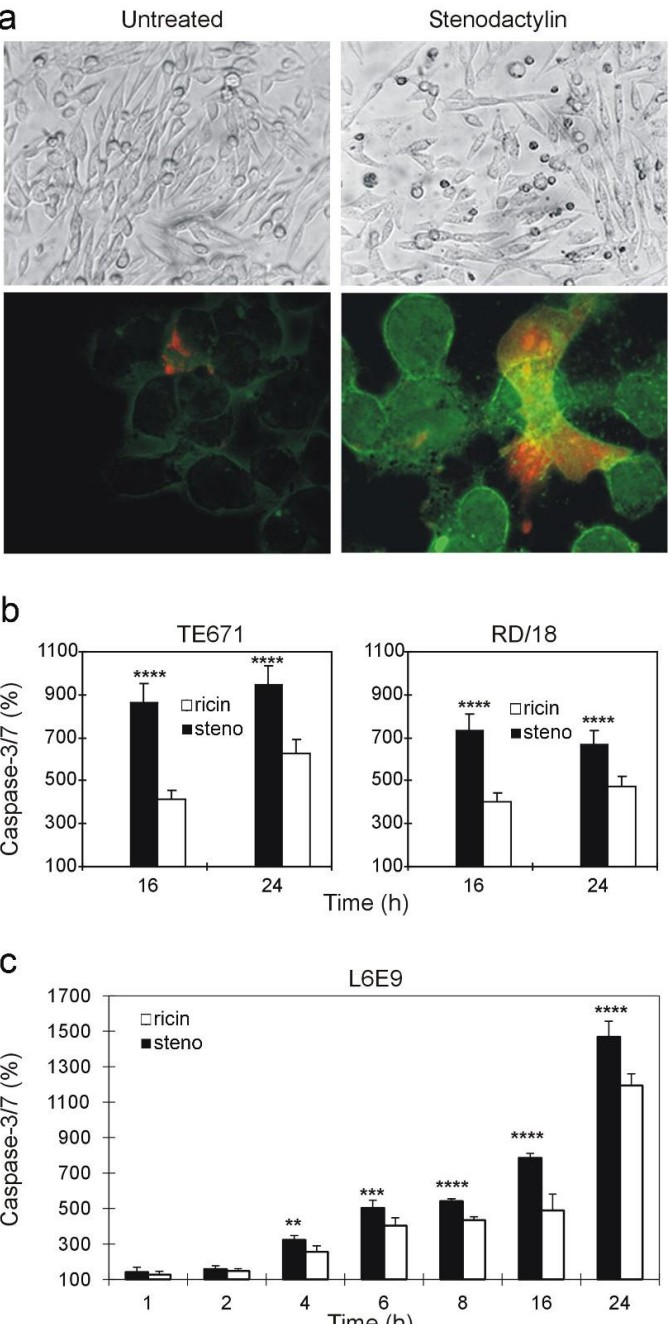

**Figure 4.** Evaluation of apoptosis by phase contrast and fluorescence microscopy (**a**). TE671 cells were treated for 24 h with 10 pM stenodactylin. Morphology was assessed by phase contrast microscopy ((**a**), **upper panels**). Apoptosis and necrosis were evaluated by fluorescence microscopy ((**a**), **lower panels**). Cells were double stained with Annexin V-FITC and PI to evaluate apoptotic (FITC+/PI− and FITC+/PI+) and necrotic cells (FITC−/PI+). Control cultures grown in the absence of RIP are also shown. Magnification 40× (phase contrast microscopy), 600× (fluorescence microscopy). Evaluation of caspase-3/7 activation in TE671 and RD/18 cells (**b**). Rhabdomyosarcoma cells were treated with 0.01 pM stenodactylin (black bars) or 10 pM ricin (white bars) and caspase activity was detected at 16 and 24 h through a luminescent assay. Evaluation of caspase-3/7 activation in L6E9 cells (**c**). Myoblasts were treated with 0.01 pM stenodactylin (black bars) or 10 pM ricin (white bars); caspase activity was detected in the time range 1–24 h. Results are means of three different experiments, each performed in duplicate. SD never exceeded 10%. Data, expressed as percentage of control values, were analyzed by the ANOVA/Bonferroni test (confidence range 95%; **** $p < 0.0001$; *** $p < 0.001$; ** $p < 0.01$).

To evaluate the involvement of apoptosis in cells treated with RIPs, caspase-3/7 activity was assayed using the minimum concentration that completely inhibited protein synthesis at 72 h (i.e., 0.01 pM for stenodactylin and 10 pM for ricin). Stenodactylin and ricin were able to activate caspase-3/7 in TE671 and RD/18 cells. As reported in Figure 4b, a massive increase in caspase activity was observed after 16 h and 24 h of incubation with both RIPs ($p < 0.0001$ RIPs versus untreated cells). The increase in caspase-3/7 was significantly greater with stenodactylin than with ricin ($p < 0.0001$ stenodactylin- versus ricin-treated cells). In order to analyze the timing of apoptotic damage in myoblast L6E9 cells, caspase-3/7 activity was assessed from 1 to 24 h. As shown in Figure 4c, both RIPs were able to significantly activate caspase-3/7 after 4 h exposure ($p < 0.01$ RIPs versus untreated cells). Stenodactylin showed greater ability than ricin to activate effector caspases; in fact, after 24 h of exposure to stenodactylin using a concentration 1000-fold lower than ricin, caspases reached values of about 1500% of controls.

### 2.5. Pulse-Chase Experiments in Target and Nontarget Cells

Pulse-chase experiments were conducted to simulate compatible conditions with an in vivo use of RIP. Cells were treated with stenodactylin from 15 to 120 min to assess the minimum required time to inhibit protein synthesis (evaluated after 72 h). In fact, longer contact times would be difficultly achievable in vivo. In TE671 cells, a contact with 10 pM stenodactylin for 120 min reduced protein synthesis to about 20%, whereas, in the same conditions, 1 pM stenodactylin reduced protein synthesis to about 75% (Figure 5a). In L6E9 cells incubated with 10 pM stenodactylin, protein synthesis was reduced to less than 25% after 15 minutes contact and completely inhibited after 60 minutes. The exposure to the RIP at 1 pM concentration caused a protein synthesis reduction to about 25% after 2 h contact (Figure 5b).

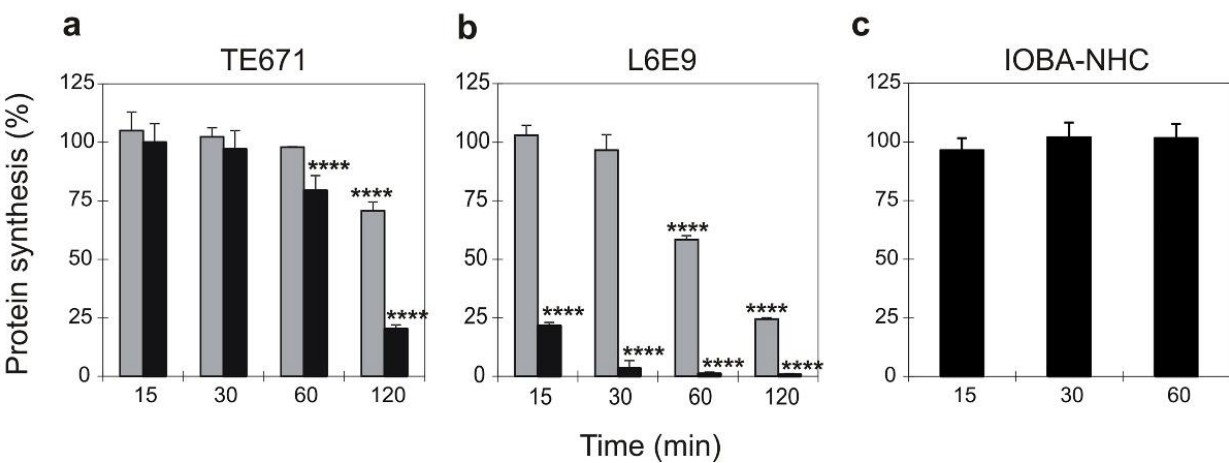

**Figure 5.** Protein synthesis inhibition evaluated through pulse-chase experiments. TE671 (**a**), L6E9 (**b**) and IOBA-NHC (**c**) cells were exposed for indicated times to stenodactylin at 10 pM (black bars) or 1 pM (grey bars) concentrations. After the indicated incubation times, stenodactylin was removed and cells were incubated for 72 h with complete medium. Results are means of two different experiments, each performed in triplicate. SD never exceeded 10%. Data were analyzed by ANOVA/Bonferroni test, followed by a comparison with Dunnett's test (confidence range 95%; **** $p \leq 0.0001$ versus untreated cells).

In a further experiment, we evaluated the possible cytotoxic effect of stenodactylin toward the tissues surrounding the treated site, in case of accidental liquid spread during the injection. We chose times largely in excess with respect to reality, as any accidental contact cannot exceed a maximum of few minutes, both because of the intraoperative irrigation of the ocular surface and because of the tear fluid dilution and eyelid movements. For this purpose, human conjunctival epithelial cell line IOBA-NHC was incubated with

10 pM of stenodactylin for 15, 30 and 60 min. The effect on protein synthesis was evaluated after 72 h. No significant cytotoxic effect was reported in the considered time range (Figure 5c).

## 3. Discussion

In this paper, two potent type 2 RIPs, namely, stenodactylin and ricin, were evaluated as candidates for experimental enzymatic surgery, as an alternative to traditional surgery and BTX for the treatment of strabismus and oculofacial dystonias. The cytotoxic effects of stenodactylin and ricin were tested on two human rhabdomyosarcoma cell lines, namely, TE671 and RD/18, and on the murine myoblast L6E9 cell line.

Stenodactylin strongly inhibited protein synthesis and the result was more toxic than ricin in all the tested cell lines. Thus, stenodactylin confirmed its high cytotoxicity similarly to other RIPs purified from the Adenia genus [28–30]. In the muscle cell lines used in these experiments, ricin showed a slightly lower cytotoxicity than that reported in the literature for other cell lines [23]. However, it should be considered that it is difficult to make a direct comparison of data available in the literature about ricin cytotoxicity because of differences in experimental approaches and technical conditions.

Both ricin and stenodactylin resulted in greater cytotoxicity toward undifferentiated cells than toward differentiated myotubes. Moreover, stenodactylin was highly more toxic on murine myoblasts than on rhabdomyosarcoma cells. Stenodactylin, at 10 pM of concentration, caused about 80% of protein synthesis inhibition after 15 min in L6E9 myoblasts and after 120 min in TE671 myocytes in pulse-chase experiments.

Our results demonstrated a prevalent involvement of apoptosis in stenodactylin-treated cells. This represents an advantage for the therapeutic use of this RIP because apoptotic death is not followed by inflammatory processes, as apoptotic cells are removed through efferocytosis by professional and nonprofessional phagocytes [31,32].

The high stenodactylin cytotoxicity would permit to reach effective dosage using very small volumes. This would allow a longer RIP persistence in the injection site and a better internalization into target cells, avoiding tear dilution.

In order to test the effects of stenodactylin in tissues surrounding the treated muscle, cytotoxicity experiments were carried out on human conjunctival epithelial cells. Stenodactylin was able to produce a cytotoxic effect on muscular cells but not on the conjunctival cells, which were exposed to stenodactylin for a time range of 15–60 min. These experimental conditions were chosen largely in excess, considering that the accidental contact of the toxin with tissues surrounding the treated muscle could not be longer than a few minutes because of the dilution affected by lachrymal washing and eyelid movement. Our results showed that even a very long exposure (60 min) to the toxin, which represents a condition that is difficult to reach in vivo, did not damage conjunctival cells. Moreover, any possible toxic effect on extramuscular tissues could be prevented by a wash with lactose-buffered solution. In fact, it was reported that in rabbit eyes, the instillation of 10% lactose solution following a massive eye administration of a type 2 RIP (1 μM) drastically reduced the toxic effects on conjunctival and corneal epithelium [33].

Stenodactylin showed $IC_{50}$ and $IC_{90}$ values of approximately 1 log and 3 logs, respectively, which are lower than those obtained with antiacetylcholine receptor (AchR)/ricin immunotoxins in the treatment of strabismus and blepharospasm [21]. Our data appear to be very encouraging when compared to those obtained with immunoconjugates. The above-reported results suggest that the therapeutic concentration could be between 1 and 10 pM. These concentrations are lower than those used in therapy with BTX [1] and mAb35/ricin (about 1000 pM for both drugs) [20]. Moreover, anti-AchR immunotoxins were effective on differentiated muscle cells (myotubes and myofibers) but were ineffective on immature muscle cells (myoblasts) that do not express AchR. For this reason, these immunotoxins were able to determine muscle degeneration but did not block regenerative phenomena. Indeed, at 14 days after inoculation of mAb35/ricin immunotoxin, rabbit extraocular muscles showed signs of regeneration [20], which continued until the restoration

of the normal number of muscular fibers after one year from the treatment [34]. Our results suggest that stenodactylin could be more suitable than immunotoxins or other substances currently used for the experimental treatment of strabismus and oculofacial dystonias.

## 4. Conclusions

Despite the results being limited only to in vitro experiments, in this paper, we demonstrate that ricin and stenodactylin can represent great candidates for in vivo locoregional treatment of strabismus and oculofacial dystonias. Furthermore, the use of these toxins and the very low effective doses may exclude the risk of cell death by necrosis and the consequent inflammatory processes, thus providing a good safety profile. In clinical application, the in vivo expected high cytotoxicity of stenodactylin on muscle cells could permit the use of very low doses of toxin and very small volumes of injection. In this way, the modulation of ablation would be facilitated and the risk of spread from the injection site could be strongly reduced. However, further studies in animal models will be needed to establish the actual in vivo efficacy of this enzymatic surgery approach.

## 5. Materials and Methods

### 5.1. Ribosome-Inactivating Proteins

The type 2 RIP stenodactylin, from the caudices of *Adenia stenodactyla*, was purified as previously described [24]. The type 2 RIP ricin was purified from the seeds of *Ricinus communis*, as described in [35].

### 5.2. Cell Lines

RIP activity was assayed on TE671 (from long-term culture of our department) and RD/18 [36] cell lines derived from human rhabdomyosarcoma, and on the L6E9 [37] cell line derived from murine myoblasts.

Cells were maintained in DMEM culture medium (Sigma-Aldrich, St. Louis, MO, USA), supplemented with 10% heat-inactivated fetal bovine serum (Sigma-Aldrich), 2 mM of L-glutamine, 100 U/mL of penicillin and 100 μg/mL of streptomycin (Sigma-Aldrich) (hereafter named complete medium), in humidified air with 5% $CO_2$ at 37 °C. Viability was checked before each experiment by trypan blue dye exclusion.

To induce differentiation, TE671 and RD/18 cell lines were maintained for 72 h in complete DMEM medium supplemented with 2% heat-inactivated horse serum (Sigma) (differentiation medium). The medium was replaced every 24 h.

IOBA-NHC cell line, derived from normal human conjunctival epithelial cells, was kindly provided by Prof. Yolanda Diebold [38], and was used as nontarget cells. The cells were cultured in DMEM/F12 (GIBCO, Carlsbad, CA, USA) supplemented with 1 μg/mL of bovine pancreas insulin, 2 ng/mL of mouse epidermal growth factor, 0.1 μg/mL of cholera toxin, 5 μg/mL of hydrocortisone, 10% fetal bovine serum, 50 U/mL of penicillin, 50 μg/mL of streptomycin and 2.5 μg/mL of amphotericin B (Sigma-Aldrich).

### 5.3. Cell Protein Synthesis

RIP cytotoxicity was evaluated by the inhibition of L-[4,5-$^3$H]leucine incorporation by cells, as described in [26]. Cells ($1 \times 10^4$/well) were incubated for 72 h with complete medium containing scalar concentrations of RIPs (from 0.01 to 100 pM). After 72 h, 0.125 μCi/well of L-[4,5-$^3$H]leucine (GE Healthcare, Buckinghamshire, UK) were added in 250 μL serum- and leucine-free medium. The radioactivity incorporated by cells was measured by a β-counter (Beckman Coulter, Fullerton, CA, USA), with Ready-Gel scintillation liquid (Beckman Coulter).

Cytotoxicity was evaluated also on differentiated cells treated with the RIPs at the same concentrations. Cells grown for one week in differentiation medium were seeded in 24-well plates ($1 \times 10^4$/well) and treated for 72 h to scalar concentrations of RIPs in 500 μL of differentiation medium. Results are means of three different experiments, each

performed in triplicate. Concentrations causing 50% ($IC_{50}$) and 90% ($IC_{90}$) inhibition of protein synthesis were calculated by linear regression.

In pulse-chase experiments, TE671 and L6E9 cells were exposed to stenodactylin at 10 or 1 pM concentrations for different incubation times (from 15 to 120 min). IOBA-NHC cells were treated with stenodactylin at 10 pM for the indicated times. After the indicated times, RIP-containing medium was removed and replaced with complete medium; after 72 h protein synthesis inhibition was evaluated as described above.

### 5.4. Cell Viability

Cell viability was evaluated with 3-(4,5-dimethylthiazol-2-yl)-5-(3-carboxymethoxyphenyl)-2-(4-sulfophenyl)-2H-tetrazolium (MTS) based colorimetric assay (CellTiter 96®Aqueous One Solution Cell Proliferation Assay, Promega, Madison, WI, USA), measuring the absorbance at 492 nm, as previously described [26]. TE671, RD/18 and L6E9 cells ($3 \times 10^3$/well) were seeded in a 96-well microtiter plate in 100 μL of complete medium, containing scalar concentrations of RIPs (from 0.01 to 100 pM).

Cell viability was evaluated also on differentiated cells treated with the RIPs at the same concentrations. Cells grown for one week in differentiation medium were seeded in 96-well plates ($3 \times 10^3$/well) and treated for 72 h to scalar concentrations of RIPs in 100 μL of differentiation medium.

Results are means of three different experiments, each performed in triplicate. The concentrations of RIP causing 50% ($LC_{50}$) and 90% ($LC_{90}$) inhibition of MTS reduction were calculated by linear regression analysis.

### 5.5. Cell Morphology Analysis

Morphological analysis of TE671 cells treated for 24 h with 10 pM of stenodactylin was conducted by phase contrast microscopy directly in 96-well plates using a digital camera from Motic Microscopes, (Xiamen, China).

### 5.6. Annexin V/Propidium Iodide (PI) Analysis

Surface exposure of phosphatidylserine in apoptotic cells was detected using Annexin V-FITC Apoptosis Detection Kit (Oncogene Science MA, USA). TE671 cells ($5 \times 10^4$/well) were seeded in 4-well microtiter plates (NUNC Roskilde, Denmark) over round glass coverslips in 250 μL of complete medium and incubated with stenodactylin at 10 pM. After 24 h of incubation, cells were washed twice with 0.14 M NaCl containing 5 mM of sodium phosphate buffer, pH 7.5 (PBS) and treated according to the manufacturer's instructions. Apoptotic cells at early stage (Annexin V+/PI−), apoptotic cells at late stage (Annexin V+/PI+) and necrotic cells (Annexin V−/PI+) were observed under a Nikon Eclipse E600W fluorescence microscope equipped with a 60× objective (Nikon, Melville, NY). Image merges were obtained by ACT-2U software (Nikon).

### 5.7. Evaluation of Caspase-3/7 Activities

The caspase-3/7 activity was assessed by the Caspase-Glo$^{TM}$ 3/7 luminescent assay (Promega). TE671, RD/18 and L6E9 cells ($2 \times 10^3$/well) were seeded in 96-well microtiter plates in 100 μL of complete medium. After 24 h, cells were incubated with 0.01 or 10 pM stenodactylin or ricin for the indicated times. After RIP incubation, the medium was removed and Caspase-Glo$^{TM}$ 3/7 Reagent was added to cells diluted 1:1 in complete medium. The luminescence was measured via Fluoroskan Ascent FL (Labsystem, Helsinki, Finland) following manufacturer's instructions.

### 5.8. Statistical Analyses

Each experiment was carried out in triplicate. Results were given as means $\pm$ SD for three different experiments. Data were analyzed by ANOVA test. Bonferroni's correction was used for multiple comparisons (confidence interval 95%). Statistical analyses were

conducted using the XLSTAT-Pro software, version 6.1.9, 2003 (Addinsoft Inc., Brooklyn, NY, USA).

**Author Contributions:** Conceptualization, A.B. and L.P.; methodology, A.B., M.B. and L.P.; validation, A.B., M.B. and L.P.; formal analysis, M.B., A.Z., L.S. and F.B.; investigation, M.B., A.Z., L.S. and F.B.; all the authors participated in writing, reviewing and editing the manuscript; funding acquisition, A.B. and L.P. All authors have read and agreed to the published version of the manuscript.

**Funding:** This study was supported by the University of Bologna with funds for selected research topics; Fondazione CARISBO, Project 2019. 0539.

**Data Availability Statement:** The data supporting the findings of this paper are available on request from the corresponding author.

**Conflicts of Interest:** The authors declare no conflict of interest.

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
