# Peer review of "Plant Toxins as Potential Alternatives to Botulinum Toxin for Eye-Movement Disorder Therapy"

_stresses, doi:10.3390/stresses3010020_

Round 1

Reviewer 1 Report

The authors aimed to determine if ribosome inactivating proteins (ricin and stenodactylin) could be used as alternatives to BTX for enzymatic surgery to correct eye movement disorders.  Well written manuscript with data clearly represented showing that RIPs are excellent candidates for this goal.  Cytotoxicity comparison between the toxins are well done.

Few minor issues:  

Lines 23, 182, 254, 262:  in vivo should be italicized

Lines 78 and 109: extra space should be removed

Line 104.  The MTS assay was used to determine viability.  If the cells become quiescent due to RIP treatment, what would you expect the MTS assay to record?  I pose the question because in our hands, albeit using different mammalian cell lines, ricin (>1 pM) generates "sick" cells but not dead based on a flow cytometric assay.

Line 197. In Figure 5C.  Only Stenodactylin is tested here.  The legend text appears to indicate that both stenodactylin and ricin were tested in all three panels.  Reword the legend to make that more clear.

Line 219.  Best to cite reference for "because apoptotic death is not followed by inflammatory processes".   Perhaps reference 15 and others?

Also, are the authors aware of the approximate half-life of ricin and stenodactylin in cell culture conditions.  If known, useful to report in this manuscript.

Overall, excellent work.  Will make a fine addition to the literature.

Author Response

The authors aimed to determine if ribosome inactivating proteins (ricin and stenodactylin) could be used as alternatives to BTX for enzymatic surgery to correct eye movement disorders.  Well written manuscript with data clearly represented showing that RIPs are excellent candidates for this goal.  Cytotoxicity comparison between the toxins are well done.

Answer: We thank the reviewer for their appreciation to our manuscript.

Few minor issues: 

Lines 23, 182, 254, 262:  in vivo should be italicized

Answer: As suggested, we italicized all “in vivo” and “in vitro” reported in the text.

Lines 78 and 109: extra space should be removed

Answer: As suggested, extra spaces have been removed.

Line 104.  The MTS assay was used to determine viability.  If the cells become quiescent due to RIP treatment, what would you expect the MTS assay to record?  I pose the question because in our hands, albeit using different mammalian cell lines, ricin (>1 pM) generates "sick" cells but not dead based on a flow cytometric assay.

Answer: In our decades lasting experience, testing various type 1 and type 2 RIPs on many cell lines, we never observed the selection of resistant cell clones or the induction of quiescence, but indeed RIPs always showed cytotoxicity even under conditions of non-logarithmic growth. This is certainly due to the ability of RIPs to deadenylate various substrates essential for cell survival, even those not related to the cell cycle. See for example: PMID: 9016590, PMID: 17434551, PMID: 26902405, and PMID: 33499082. Depending on the cell line used, some cells could appear sick after 24 hours of intoxication, even with ricin >1 pM. Anyway, in our experience no living cell can be longer detected after 72 hours.

Line 197. In Figure 5C.  Only Stenodactylin is tested here.  The legend text appears to indicate that both stenodactylin and ricin were tested in all three panels.  Reword the legend to make that more clear.

Answer: We apologize for the mistake. We have corrected the legend to Figure 5C accordingly to reviewer’s suggestion.

Line 219 (now 229)  Best to cite reference for "because apoptotic death is not followed by inflammatory processes".   Perhaps reference 15 and others?

Answer: It is well known that apoptosis, unlike necrosis, does not cause inflammation in vivo since apoptotic cells and apoptotic bodies do not show membrane alterations and, consequently, no DAMPs are released. In line 229 we have better specified this concept and two new references have been added.

Also, are the authors aware of the approximate half-life of ricin and stenodactylin in cell culture conditions.  If known, useful to report in this manuscript.

Answer: We have not available data about the half-life of ricin and stenodactylin in the experimental model tested in this work. However, the differences between ricin- and stenodactylin- cytotoxicity observed in our experiments could be due to a higher ricin inactivation with respect to stenodactylin, but could depend also on a number of other factors, such as cell binding, intracellular routing, degradation, exocytosis and so on.

Overall, excellent work.  Will make a fine addition to the literature.

Answer: We thank the reviewer for their appreciation to our paper.

Reviewer 2 Report

Intramuscular injection of botulinum toxin (BTX) is an alternative to surgery for the treatment of ocular muscle spasm. However, its effect is not permanent. In this manuscript, the authors propose the use of two toxins, ricin and stenodactylin, as substitutes for botulinum toxin in the treatment of strabismus and oculofacial dystonias. Ricin and stenodactylin are two type 2 ribosome-inactivating proteins (RIPs). Type 2 RIPs consist of an enzymatically active A chain linked via a disulfide bond to a B chain with lectin properties. The B chain has a strong affinity for cell surface sugars and can facilitate the entry of toxins into cells, making many type 2 RIPs highly toxic to cells and animals.

The authors study the toxicity of these type 2 RIPs in two human rhabdomyosarcoma cell lines (TE671 and RD/18), in a murine myoblast cell line (L6E9), and in a human conjunctival epithelial cell line (IOBA-NHC), finding several results that are really encouraging:

1.- Ricin and stenodactylin inhibit protein synthesis and are toxic to TE671 and RD/18 cells in the pM range.

2.- Stenodactylin is 10 times more toxic than ricin for TE671 and RD/18 cells.

3.- Stenodactylin is much more toxic for L6E9 cells (sub-pM range)

4.- Stenodactylin and ricin induce apoptosis in TE671, RD/18 and L6E9 cells.

5.- Stenodactylin induces apoptosis in TE671 and RD/18 cells in the pM range and in L6E9 cells in the sub-pM range.

6.- Stenodactylin does not inhibit protein synthesis in IOBA-NHC cells at a concentration 10 times higher than that which inhibits protein synthesis in TE671 and L6E9 cells.

These results indicate that ricin, and especially stenodactylin, may be promising candidates to replace botulinum toxin in the treatment of strabismus and oculofacial dystonias.

On the other hand, the subject addressed in this manuscript is worthy of investigation and the conclusions are supported by the experimental data. This study will stimulate further research on this field. On this basis, the paper deserves publication in the present form.

Minor points:

Page 1, lines 29-30: better in alphabetical order?

Page 3, Figure 1b: Indicate the type of cells in the second panel

Page 4, Figure 2b: Indicate the type of cells in the second panel

Page 5, Figure 3c: indicate the units

Page 5, line 155: Change “The caspase-3/7 increase” by “The increase of caspase 3/7 activity”

Page 6, lines 171-173: Change “black” and “white” by “black bars” and “white bars”

Page 9, line 306: MTS (3-(4,5-dimethylthiazol-2-yl)-5-(3-carboxymethoxyphenyl)-2-(4-sulfophenyl)-2H-tetrazolium)

Pages 11-12: correct double numbering of references

Author Response

Intramuscular injection of botulinum toxin (BTX) is an alternative to surgery for the treatment of ocular muscle spasm. However, its effect is not permanent. In this manuscript, the authors propose the use of two toxins, ricin and stenodactylin, as substitutes for botulinum toxin in the treatment of strabismus and oculofacial dystonias. Ricin and stenodactylin are two type 2 ribosome-inactivating proteins (RIPs). Type 2 RIPs consist of an enzymatically active A chain linked via a disulfide bond to a B chain with lectin properties. The B chain has a strong affinity for cell surface sugars and can facilitate the entry of toxins into cells, making many type 2 RIPs highly toxic to cells and animals.

The authors study the toxicity of these type 2 RIPs in two human rhabdomyosarcoma cell lines (TE671 and RD/18), in a murine myoblast cell line (L6E9), and in a human conjunctival epithelial cell line (IOBA-NHC), finding several results that are really encouraging:

1.- Ricin and stenodactylin inhibit protein synthesis and are toxic to TE671 and RD/18 cells in the pM range.

2.- Stenodactylin is 10 times more toxic than ricin for TE671 and RD/18 cells.

3.- Stenodactylin is much more toxic for L6E9 cells (sub-pM range)

4.- Stenodactylin and ricin induce apoptosis in TE671, RD/18 and L6E9 cells.

5.- Stenodactylin induces apoptosis in TE671 and RD/18 cells in the pM range and in L6E9 cells in the sub-pM range.

6.- Stenodactylin does not inhibit protein synthesis in IOBA-NHC cells at a concentration 10 times higher than that which inhibits protein synthesis in TE671 and L6E9 cells.

These results indicate that ricin, and especially stenodactylin, may be promising candidates to replace botulinum toxin in the treatment of strabismus and oculofacial dystonias.

On the other hand, the subject addressed in this manuscript is worthy of investigation and the conclusions are supported by the experimental data. This study will stimulate further research on this field. On this basis, the paper deserves publication in the present form.

Answer: We thank the reviewer for their appreciation to our paper.

Minor points:

Page 1, lines 29-30: better in alphabetical order?

Answer: The keywords have been sorted in alphabetical order.

Page 3, Figure 1b: Indicate the type of cells in the second panel

Answer: Figure 1b has been corrected, indicating the cell line name in the second panel.

Page 4, Figure 2b: Indicate the type of cells in the second panel

Answer: Figure 2b has been corrected, indicating the cell line name in the second panel.

Page 5, Figure 3c: indicate the units

Answer: Figure 3c has been corrected, indicating the units (pM).

Page 5, line 155: Change “The caspase-3/7 increase” by “The increase of caspase 3/7 activity”

Page 6, lines 171-173: Change “black” and “white” by “black bars” and “white bars”

Page 9, line 306: MTS (3-(4,5-dimethylthiazol-2-yl)-5-(3-carboxymethoxyphenyl)-2-(4-sulfophenyl)-2H-tetrazolium)

Answer: All the suggestions have been accepted and the text has been accordingly modified.

Pages 11-12: correct double numbering of references

Answer: As requested, the double numbering of references has been removed.

Reviewer 3 Report

An interesting idea:  and the first step is, of course, examining the sensitivities of target and non-target cells to the toxins.

I do have a couple of comments, though.

A 72h exposure to toxins is appropriate for cell viability studies, but does not really give a measure of reduced protein synthesis because cell viability is conflated in these experiments.  Protein synthesis experiments should be performed in the time period before apoptotic loss of cells.

Are these rhabdomyosarcoma cell lines relatively resistant to toxin?   Ricin has an IC50 of 34 pM to TE671 cells (Fig 1) - approx 2 ng/ml after a 72 h exposure.  To HeLa cells for example, the IC50 is about 1 ng/ml for a brief (6h) exposure.

It is clear that  L6E9 cells respond rapidly to toxin treatment (Fig 5) and TE671 are rather less sensitive:  but it is hard to judge the sensitivity of non-target IOBA-NHC cells because they were treated to a maximum of 60 min, rathe than 120 minutes. This panel of data (Fig 5c) should be extended to 129 minutes, so that direct comparisons can be made.

Author Response

An interesting idea:  and the first step is, of course, examining the sensitivities of target and non-target cells to the toxins.

I do have a couple of comments, though.

A 72h exposure to toxins is appropriate for cell viability studies, but does not really give a measure of reduced protein synthesis because cell viability is conflated in these experiments.  Protein synthesis experiments should be performed in the time period before apoptotic loss of cells.

Answer: In our model, the protein synthesis experiments carried out at long times represent an index of the reduction of proliferation in treated cells respect to the control cells. These experiments can give a useful confirmation to the cytotoxicity/viability experiments and all together they enforce the evidence of the cytoxic effect of the tested RIPs. It is well known that RIP toxicity is based on de-adenylation of several substrates in addition to rRNA. Furthermore, some research groups have demonstrated that apoptosis is induced before protein synthesis inhibition. From this point of view, the inhibition of protein synthesis at short incubation times would not provide fundamental information.

Are these rhabdomyosarcoma cell lines relatively resistant to toxin?   Ricin has an IC50 of 34 pM to TE671 cells (Fig 1) - approx 2 ng/ml after a 72 h exposure.  To HeLa cells for example, the IC50 is about 1 ng/ml for a brief (6h) exposure.

Answer: In our experience, the IC50s can be influenced by the experimental conditions and it is quite difficult to strictly compare data from different experiments and methods. Accordingly, we added a sentence in the discussion to better specify this concept (lines 217-221 page 10): “In the muscle cell lines used in these experiments, ricin showed a slight lower cytotoxicity than that reported in literature for other cell lines [23]. But, it should be considered that it is difficult to make a direct comparison of data available in literature about ricin cytotoxicity, because of differences in experimental approaches and technical conditions.”

It is clear that  L6E9 cells respond rapidly to toxin treatment (Fig 5) and TE671 are rather less sensitive:  but it is hard to judge the sensitivity of non-target IOBA-NHC cells because they were treated to a maximum of 60 min, rather than 120 minutes. This panel of data (Fig 5c) should be extended to 129 minutes, so that direct comparisons can be made.

Answer: We thank the reviewer for their careful observation. The experiments conducted on the non-target cells were carried out trying to mimic the condition that could occur in vivo during the treatment as a consequence of accidental toxin leakage and contact with the tissue surrounding the treated site. In our opinion, the duration of an accidental contact cannot exceed a maximum of few minutes, both because of the intraoperative irrigation of the ocular surface and because of the tear fluid dilution and eyelid movements. For all these reasons, the experimental conditions we chose (toxin dose of 10 pM and contact times of 15-60 minutes) are already very unlikely to be achieved in vivo. We have clarified this concept, modifying the text in Results 2.5 - Lines 194-196 and in Discussion - Lines 237-242.

Round 2

Reviewer 3 Report

Authors have responded